# Isokinetic Profile of Elite Serbian Female Judoists

**DOI:** 10.3390/ijerph18136988

**Published:** 2021-06-29

**Authors:** Wieslaw Blach, Miodrag Drapsin, Nemanja Lakicevic, Antonino Bianco, Tamara Gavrilovic, Roberto Roklicer, Tatjana Trivic, Ognjen Cvjeticanin, Patrik Drid, Maciej Kostrzewa

**Affiliations:** 1Faculty of Physical Education & Sport, University School of Physical Education in Wroclaw, 51-612 Wroclaw, Poland; wieslaw.judo@wp.pl; 2Medical Faculty, University of Novi Sad, 21000 Novi Sad, Serbia; miodrag.drapsin@mf.uns.ac.rs; 3Sport and Exercise Sciences Research Unit, University of Palermo, 90133 Palermo, Italy; lakinem89@gmail.com (N.L.); antonino.bianco@unipa.it (A.B.); 4Serbian Institute of Sport and Sports Medicine, 11000 Belgrade, Serbia; tamara.gavrilovic@rzsport.gov.rs; 5Faculty of Sport and Physical Education, University of Novi Sad, 21000 Novi Sad, Serbia; roklicer.r@gmail.com (R.R.); ttrivic@yahoo.com (T.T.); ognjencvjeticanin@gmail.com (O.C.); patrikdrid@gmail.com (P.D.); 6Institute of Sport Science, The Jerzy Kukuczka Academy of Physical Education in Katowice, 40-065 Katowice, Poland

**Keywords:** isokinetic strength, quadriceps, hamstring, combat sports, martial arts

## Abstract

Elite judo athletes undergo vigorous training to achieve outstanding results. In pursuit of achieving competitive success, the occurrence of injuries amongst judo athletes is not rare. The study aimed to perform a knee flexors and extensors isokinetic torque analysis in elite female judo athletes. Fifty-eight elite female judo athletes of the Serbian national team (21.02 ± 3.11 years; 62.36 ± 11.91 kg, 165.04 ± 10.24 cm, training experience 12.72 ± 2.98 years) volunteered to participate in this study. The range of motion (ROM) was set at 90°. Testing was performed in a concentric-concentric mode for the testing speed of 60 °/s. Five maximal voluntary contractions of knee extensors and knee flexors muscle groups were measured for both legs. The obtained data showed a statistically significant difference in absolute torque values among different categories as heavier athletes demonstrated higher values. Post hoc analysis showed a significant difference between weight categories, as heavier athletes demonstrated higher values, while no significant differences in normalized torque values for different weight categories were observed. The implementation of new elements and training modalities may improve performance and prevent lateral asymmetry, thus reducing the risk of injury.

## 1. Introduction

To excel in their careers and achieve outstanding results, athletes have to assess their performance frequently. If done correctly and in a timely manner, this procedure allows athletes and coaches to identify potential flaws in their training regimen, which can consequently upgrade their performance and improve their career trajectories [1]. In addition, properly prescribed training should not only contribute to better performance but also injury prevention [2]. Skill-related physical fitness elements such as speed, balance, agility, coordination, and reaction time are essential aspects of performance [3], especially at the elite level [4,5,6]. Even though fitness can be genetically determined to a degree, it can also be greatly affected by a training program that further determines the fitness level of an individual [7]. Innovative recovery systems also have an impact on the athletic performance of combat fighters [8]. It is not a surprise then that some athletes spend 17% of their waking time in training [9]. Exceptional engagement in training might be particularly important for judo since it is a very demanding type of physical activity that requires the involvement of the entire body and a modifiable set of technical-tactical skills [10,11]. However, high training volumes do not necessarily lead to better results, and therefore it is important for athletes to train smarter and not inevitably harder. Isokinetic torque analysis allows researchers to assess dynamic muscular force at a constant speed with a device that provides variable resistance whereby force generated by the muscle is constantly matched [12]. Herewith, force generation over the full range of motion is depicted. Isokinetic dynamometers have been shown to be highly reliable [13], especially when testing uniaxial joints. Through isokinetic analysis, key variables such as peak torque and mean peak torque for different speed velocities represented in °/s can be extracted to evaluate muscle performance [12]. Furthermore, isokinetic testing provides valuable information about the asymmetry in strength between muscle groups that can be a potential site for injury, but more importantly, for the injury prevention program. Knowing that females are more prone to suffer from anterior cruciate ligament injury (ACL) [14], which is common in judo [15], it is of great importance to access lower limb muscle strength in female judokas. The study was conducted to additionally develop classificatory standards in peak isokinetic muscle torques of the knee joint extensors and flexors for female judo athletes. The aim of this study was to perform lower limb isokinetic analysis in elite Serbian female judokas. We hypothesized that there is a significant difference between female judokas of different weight categories in the peak isokinetic muscle torques of the knee joint extensors and flexors.

## 2. Materials and Methods

### 2.1. Sample

Fifty-eight elite female judo athletes of the Serbian national team (21.02 ± 3.11 years; 62.36 ± 11.91 kg, 165.04 ± 10.24 cm, training experience 12.72 ± 2.98 years) volunteered to participate in this study. All judokas were divided into 7 category groups established by the International Judo Federation (IJF) (−48 kg, *n* = 8; −52 kg, *n* = 8; −57 kg, *n* = 8; −63 kg, *n* = 9; −70 kg, *n* = 8; −78 kg, *n* = 8; +78 kg, *n* = 9). To be included in the present study, athletes had to compete in an international level competition as a member of a Serbian national team and to have a minimum black belt. Participants were excluded if they had less than seven years of experience in judo and they had to be free of injury at the time of testing and implementing their regular training regimen. All participants were thoroughly informed about the research design and provided written consent to participate in the study. This study was approved by the Institutional Review Committee of the University of Novi Sad (Ref. No. 46-06-02/2020-1) and was conducted under the Declaration of Helsinki.

### 2.2. Experimental Procedure and Data Analysis

Test familiarization and warm-up sessions for 15 min were performed. Whereby each athlete completed five trials on the isokinetic dynamometer (Humac Norm, Lumex, Ronkonkoma, NY, USA). Before each testing, the device was calibrated. The range of motion (ROM) was set at 90°. Judokas were allowed to hold the seat handles and were instructed to quit in case of emergency. The seat was adjusted according to the weight and height of the participants, who had to have their hip joint in a perpendicular position. The straps were placed over both shoulders, and the upper part of the thigh was also stabilized with a strap to ensure stabilization and minimize substitution. Testing was performed in a concentric-concentric mode for the testing speed of 60 °/s; five maximal voluntary contractions of quadriceps (knee extensors) and hamstring (knee flexors) muscle groups were measured for both legs. The test velocity was used according to previous studies [6,16].

All measurements were conducted by the same researcher under the same protocol. Measured parameters of peak torque (in Nm) for knee flexors and extensors were recorded. The hamstring to quadriceps ratio was calculated by dividing the peak torque of the hamstring to the quadriceps muscle group of the same leg.

### 2.3. Statistical Analysis

IBM SPSS Statistics v23.0 (IBM Inc., Armonk, NY, USA) was used for statistical analysis. Normality of distribution was confirmed by the Shapiro-Wilk test, while equality of variances was assessed using Levene’s test. In order to compare different weight categories of female judoists, a one-way analysis of variance with Tukey’s post hoc analysis was used. Effect size (η^2^) was calculated as well and defined 0.2, 0.5, and 0.8 as small, medium, and large. Pearson’s correlation was calculated between training experience and torque values. Statistical significance was set at *p* < 0.05.

## 3. Results

There were statistically significant differences between body height and body weight between different weight classes (Table 1). No significant differences were noted in terms of age and years of experience.

The obtained data showed a statistically significant difference in absolute torque values among different categories ± standard deviation (Table 2). Likewise, no significant differences were noted in the hamstring to quadriceps ratio between judo athletes.

Post hoc analysis showed that a significant difference between weight categories, as heavier athletes, demonstrated higher values.

The right/left difference (i.e., higher value-lower value) between the torque of knee extensors or flexors is less than 10% (respectively, 6%, 8%). There are no statistically significant differences between left and right leg extensors (F = 0.577, *p* = 0.747) or flexors (F = 0.879, *p* = 0.521) between different weight categories.

However, higher values have been achieved in extension rather than flexion mean torque (ANOVA, F = 17.200, *p* < 0.001 and F = 3.800, *p* = 0.010, respectively) (Figure 1). The post hoc analysis did not show that significant differences in normalized torque values for different weight categories were observed.

Similar results were observed for the left leg. By comparing the extension and flexion movement, a statistically significant difference in absolute torque values among different categories was noted (ANOVA, F = 7.700, *p* < 0.001 and F = 3.400, *p* = 0.015, respectively) (Figure 2). The post hoc analysis did not show a significant difference in normalized torque values for different weight categories.

A moderate negative correlation was observed for the right and the left leg flexors in terms of training experience (*r* = −0.33, *p* = 0.010; *r* = −0.51, *p* < 0.001, respectively) (Figure 3).

A moderate negative correlation was observed in the hamstring to quadriceps ratio of right and left leg relating to training experience (*r* = −0.4, *p* = 0.02; *r* = −0.57, *p* < 0.001, respectively) (Figure 4).

## 4. Discussion

The aim of the present study was to perform an isokinetic analysis of hamstring and knee muscles in elite female judo athletes. Acquired results revealed that heavier athletes demonstrated higher isokinetic muscle torque. The results in our previous study showed that elite judokas produced higher peak isokinetic muscle torque values while testing the speed of 60 °/s for thigh muscles than sub-elite male judokas [6]. However, previous studies on isokinetic performance on judo athletes have shown that the heavier the athletes (with higher fat mass), the higher the risk for ACL rupture, which pertains even more so to female athletes [16]. Indeed, post-pubescent female athletes are at a 3.5 times higher risk of developing ACL injury when compared to their male counterparts [17,18,19]. A definitive answer to the question of why females are more susceptible to ACL injuries is currently unknown, but several factors such as fluctuation in sex hormones during the menstrual cycle; sex differences in knee anatomy; and dynamic neuromuscular imbalances have been proposed as theories [19]. Furthermore, with respect to muscle strength, females display less quadriceps and hamstring strength even when normalizing for body mass [20,21]. This notion is very significant because several studies have found that strengthening exercises focused on the quadriceps and hamstring areas are knee-protective [22,23,24,25].

Judo is a very demanding type of sport that requires the involvement of the entire body and a modifiable set of technical and tactical skills [9]. Judo athletes often engage in 20 h of intense training per week [26], which might result in the development of overtraining syndrome [11] and further leads to an elevated risk of developing an injury. Pocecco et al. [27] showed that during the Olympic Games in 2008 and 2012, the average injury risk was about 11–12% of judo athletes. Studies have shown that even when athletes attempt to come back to the sport after ACL reconstruction, no standardized isokinetic protocol following ACL reconstruction has been developed, whereas isokinetic torque measures discussed in a systematic review by Undheim et al. [28] have not been validated as useful predictors of a successful return to sport. Lower-limb muscle groups are particularly important in judo. Hence, we suggest paying particular attention to developing the strength of the muscles of the lower extremities in the training process, which is related to the health aspect of judo practitioners [29]. Certainly, more studies are needed in the field of isokinetic analysis in professional sports. Isokinetic testing can provide valuable information about the strength of individual muscle groups and identify potential imbalance between them [30] where new elements of training and various training modalities might be essential to improving performance and preventing injuries due to lateral asymmetry [31]. Indeed, a recent study by Kons et al. [32] showed that lower-limb muscle power was positively associated with judo-specific performance. Results in our study indicate that elite Serbian female judokas do not have an elevated risk of injury due to lateral or unilateral asymmetry between knee flexors and extensors in isokinetic muscle torques values. Values of bilateral asymmetry were lower than 10%. Unilateral asymmetry values were in the range of 50 to 60% between knee flexors and extensors. Our results are in line with previous studies [31,33].

We recognize that the relatively small number of participants by weight categories is a limiting factor of the study. Nevertheless, the strength of this study was the sample of Serbian elite-level female judo athletes regularly performing in international competitions.

## 5. Conclusions

Judo coaches and scientists can use isokinetic profiles regarding the strength of certain muscle groups and early detection of imbalances between muscle groups in judo. Therefore, we recommend the inclusion of isokinetic dynamometry as an injury prevention and strength diagnostic tool for elite judokas. The results of this study are presented using weight categories that could be important for coaches and athletes to control performance status and to have an indication of which aspect needs to be improved. Accordingly, methods to improve strength should be carefully selected and adopted in alignment with obtained isokinetic data.

## Figures and Tables

**Figure 1 ijerph-18-06988-f001:**
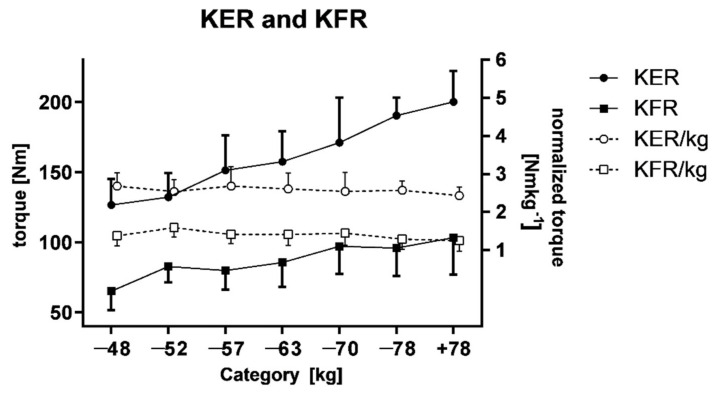
Mean torque and normalized torque values of extensors and flexors of the right knee.

**Figure 2 ijerph-18-06988-f002:**
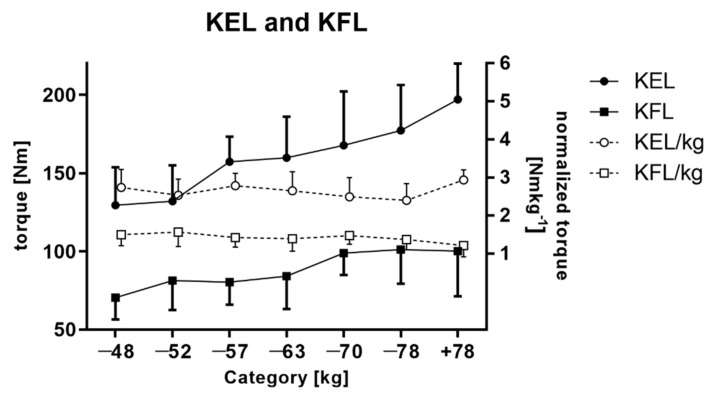
Mean torque and normalized torque values of extensors and flexors of the left knee.

**Figure 3 ijerph-18-06988-f003:**
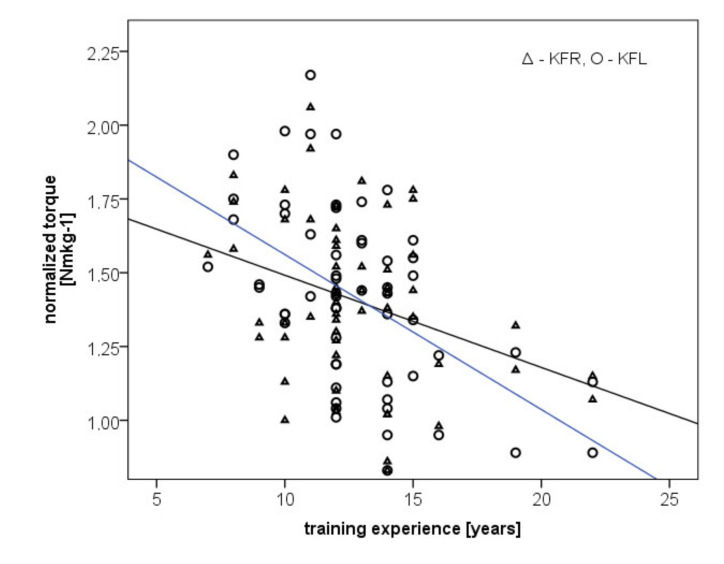
Correlation between normalized torque values of left and right leg flexors and training experience in years.

**Figure 4 ijerph-18-06988-f004:**
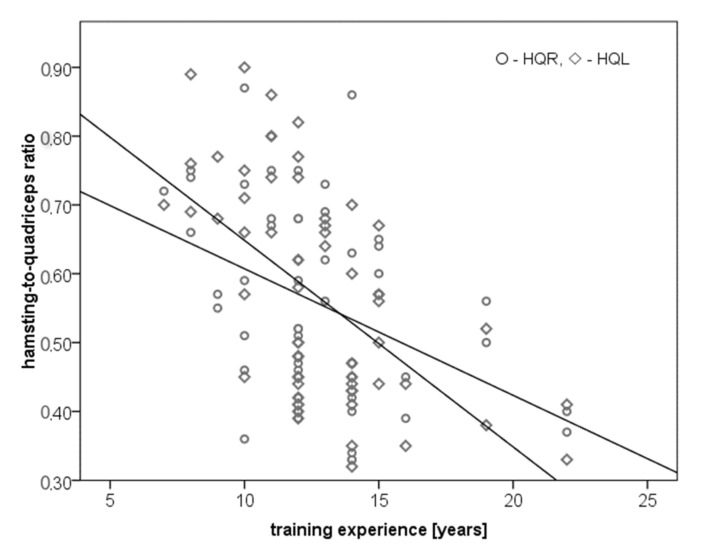
Correlation between hamstring to quadriceps ratio and training experience in years.

**Table 1 ijerph-18-06988-t001:** Physical characteristics of the participants.

Variable	−48 kg (*n* = 8)	−52 kg (*n* = 8)	−57 kg (*n* = 8)	−63 kg (*n* = 9)	−70 kg (*n* = 8)	−78 kg (*n* = 8)	+78 kg (*n* = 9)	Statistics
Height (cm)	148.50 ± 1.61	160.00 ± 2.41	163.57 ± 4.87 ^a^	166.12 ± 4.54 ^a^	169.57 ± 11.22 ^a^	172.10 ± 8.77 ^a^	174.54 ± 6.56 ^a^	F = 5.42, *p* = 0.001, η^2^ = 0.522
Weight (kg)	47.12 ± 1.80	52.00 ± 1.80 ^a^	56.62 ± 2.06 ^a, b^	60.55 ± 1.87 ^a, b, c^	67.25 ± 1.98 ^a, b, c, d^	74.16 ± 3.12 ^a, b, c, d, e^	82.11 ± 4.25 ^a, b, c, d, e, f^	F = 204.75, *p* = 0.000, η^2^ = 0.960
Age (y)	20.62 ± 1.92	19.90 ± 3.24	21.25 ± 2.96	22.77 ± 4.99	19.75 ± 2.65	21.00 ± 2.45	21.01 ± 3.10	F = 1.047, *p* = 0.410, η^2^ = 0.109
Training experience (y)	12.12 ± 2.53	12.10 ± 4.28	13.25 ± 1.38	14.88 ± 5.56	11.12 ± 1.12	12.16 ± 1.94	13.11 ± 1.36	F = 1.470, *p* = 0.207, η^2^ = 0.147

Significantly different from: ^a^ –48 kg; ^b^ –52 kg; ^c^ –57 kg; ^d^ –63 kg; ^e^ –70 kg; ^f^ –78 kg.

**Table 2 ijerph-18-06988-t002:** Differences between weight categories of female judokas in peak muscle torques of the knee joint extensors and flexors.

Variable	−48 kg (*n* = 8)	−52 kg (*n* = 8)	−57 kg (*n* = 8)	−63 kg (*n* = 9)	−70 kg (*n* = 8)	−78 kg (*n* = 8)	+78 kg (*n* = 9)	Statistics
**Mean Torque**								
KER (Nm)	126.50 ± 18.36	132.00 ± 17.13	151.25 ± 24.92	157.33 ± 21.63	171.00 ± 32.08 ^a, b^	190.33 ± 12.78 ^a, b, c^	200.11 ± 22.03 ^a, b, c, d^	F = 13.074, *p* = 0.000, η^2^ = 0.33
KEL (Nm)	129.37 ± 24.28	131.90 ± 22.98	157.12 ± 16.00	159.66 ± 26.19	167.62 ± 34.47	177.16 ± 29.06 ^a, b^	197.00 ± 22.94 ^a, b, c, d^	F = 7.738, *p* = 0.000, η^2^ = 0.27
KFR (Nm)	65.12 ± 13.46	82.70 ± 11.48	79.87 ± 13.72	85.44 ± 17.57	97.12 ± 19.65 ^a^	95.83 ± 20.03 ^a^	102.88 ± 26.63 ^a^	F = 4.088, *p* = 0.002, η^2^ = 0.53
KFL (Nm)	70.37 ± 13.93	81.20 ± 18.83	80.25 ± 14.42	84.00 ± 20.88	98.75 ± 13.76	101.33 ± 21.92	101.22 ± 28.30 ^a^	F = 3.086, *p* = 0.012, η^2^ = 0.24
**Normalized Torque**								
KER/kg (Nmkg^−1^)	2.68 ± 0.35	2.53 ± 0.31	2.68 ± 0.49	2.60 ± 0.41	2.54 ± 0.49	2.57 ± 0.24	2.43 ± 0.22	F = 0.458, *p* = 0.836, η^2^ = 0.051
KEL/kg (Nmkg^−1^)	2.74 ± 0.47	2.53 ± 0.42	2.78 ± 0.32	2.64 ± 0.50	2.49 ± 0.50	2.39 ± 0.43	2.39 ± 0.26	F = 1.054, *p* = 0.402, η^2^ = 0.110
KFR/kg (Nmkg^−1^)	1.37 ± 0.26	1.59 ± 0.24	1.41 ± 0.23	1.41 ± 0.29	1.44 ± 0.31	1.29 ± 0.27	1.24 ± 0.27	F = 1.535, *p* = 0.186, η^2^ = 0.153
KFL/kg (Nmkg^−1^)	1.49 ± 0.29	1.56 ± 0.38	1.41 ± 0.24	1.38 ± 0.32	1.47 ± 0.22	1.36 ± 0.27	1.22 ± 0.29	F = 1.198, *p* = 0.323, η^2^ = 0.124
**Hamstring to quadriceps ratio**								
HQR (%)	0.52 ± 0.14	063 ± 0.10	0.54 ± 0.17	0.55 ± 0.16	0.58 ± 0.13	0.50 ± 0.12	0.51 ± 0.11	F = 0.876, *p* = 0.519, η^2^ = 0.093
HQL (%)	0.56 ± 0.15	0.62 ± 0.14	0.51 ± 0.13	0.54 ± 0.20	0.61 ± 0.16	0.58 ± 0.17	0.51 ± 0.14	F = 0.611, *p* = 0.720, η^2^ = 0.067

KER—knee extension right; KEL—knee extension left; KFR—knee flexion right; KFL—knee flexion left; KER/kg—knee extension right related to weight; KEL/kg—knee extension left related to weight; KFR/kg—knee flexion right related to weight; KFL/kg—knee flexion left related to weight; HQR—hamstring to quadriceps ratio of the right leg, HQL—hamstring to quadriceps ratio of the left leg. Significantly different from: ^a^ –48 kg; ^b^ –52 kg; ^c^ –57 kg; ^d^ –63 kg.

## Data Availability

The dataset used and/or analyzed during the current study is available from the corresponding author in response to a reasonable request. Due to patient’s data, privacy data are not made available publicly.

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
