# Peer review of "Isokinetic Profile of Elite Serbian Female Judoists"

_ijerph, 2021, doi:10.3390/ijerph18136988_

Round 1
Reviewer 1 Report
The aim of the study was to perform a quadriceps and hamstring isokinetic analysis in elite female judo athletes. Although this is a great study, a low sample size difficult the concrete conclusions.
Additionally, as weakness of manuscript, we highlighted the lack of information regarding to inclusion and exclusion criteria. What is the name and number of ethical committee? In addition, what is the execution time of exercise (time between series/exercise)?
The major concern: Why to do assess of peak torque, knee flexors and extensors (quadriceps)? The muscles of the lower limb are not more precise than higher limbs. Moreover, the isokinetic strength measures are not validated for this population.
As strengths: I highlighted the evaluation of athletes and not recreational subjects.
-in the methods: more details and information are required.
-All data presented in the figures must be normalized by body weight, since heavier people had higher muscle strength than thin subjects.
-The discussion is very succinct, and require more discussion. In addition, no strengths and limitations were described. The lines 179-180 are unnecessary.
Author Response
R1
The aim of the study was to perform a quadriceps and hamstring isokinetic analysis in elite female judo athletes. Although this is a great study, a low sample size difficult the concrete conclusions.
The sample consisted of international level Serbian female judo athletes.
Additionally, as weakness of manuscript, we highlighted the lack of information regarding to inclusion and exclusion criteria. What is the name and number of ethical committee? In addition, what is the execution time of exercise (time between series/exercise)?
Thank you for the suggestion. Ethics approval has been added to the methods section. For an athlete to take part in our study, they had to be involved in one of the international level competition as a member of a Serbian national team and to have a minimum black belt. Participants were excluded if they had less than seven years of experience in judo and those who reported some sort of musculoskeletal injury within the last year.
The major concern: Why to do assess of peak torque, knee flexors and extensors (quadriceps)? The muscles of the lower limb are not more precise than higher limbs. Moreover, the isokinetic strength measures are not validated for this population.
As knee injury is one of the most common injuries in this particular population, the aim of the study was aimed at the most important muscle groups of the knee. Since very few studies have been published on this topic, the aim of the study was also to create an isokinetic profile of female judo athletes in various weight classes.
As strengths: I highlighted the evaluation of athletes and not recreational subjects.
-in the methods: more details and information are required.
Thank you for the suggestion. This section has been expanded.
-All data presented in the figures must be normalized by body weight, since heavier people had higher muscle strength than thin subjects.
Please notice in Figures 1 and 2 normalized values (shown in a dotted line).
-The Discussion is very succinct, and require more Discussion. In addition, no strengths and limitations were described. The lines 179-180 are unnecessary.
A paragraph on the limitations has been included, and lines 179-180 have been excluded.
Reviewer 2 Report
With this study authors aimed to analyze quadriceps and hamstring isokinetic strength in judo female elite athletes
The introduction should clarify the rationale for performing the study and enlighten in what way the quadriceps and hamstring strength can contribute to judo performance or injury prevention. It should also approach what is known about the type of contraction (concentric eccentric isometric) and the speed that best assess the strength.
Please describe the study purposes according to the statistic performed.
Methods: explain how isokinetic data was processed.
Discussion should focus on the results obtained in your study taking into account the study aim.
The current discussion approaches several items not studied by the authors, like gender strength differences, ACL, and other judo injuries. The reason why authors approach them must be clear. If authors want to explore the injuries it must be clear what is known about the strength influence in their occurrence, and studies that approached it.
On contrary, the obtained results are not enough discussed nor the study limitations referred. Please explain your results and compare them with those obtained by other authors.
The conclusion must resume your study findings. The statement “Judo movement patterns are extremely demanding on the thigh muscles” is not a conclusion from your study that must be corrected.
Author Response
R2
With this study authors aimed to analyze quadriceps and hamstring isokinetic strength in judo female elite athletes
The introduction should clarify the rationale for performing the study and enlighten in what way the quadriceps and hamstring strength can contribute to judo performance or injury prevention. It should also approach what is known about the type of contraction (concentric eccentric isometric) and the speed that best assess the strength.
The introduction has been extended, and it now clearly describes the significance of selected muscle groups in judo training. Also, a detailed explanation has been added whereby we explained isokinetic testing at a particular angular speed.
Please describe the study purposes according to the statistic performed.
The aim of the study has been adjusted. Statistical procedures have also been adjusted accordingly.
Methods: explain how isokinetic data was processed.
Corrected.
Discussion should focus on the results obtained in your study taking into account the study aim.
The discussion section has been improved.
The current Discussion approaches several items not studied by the authors, like gender strength differences, ACL, and other judo injuries. The reason why authors approach them must be clear. If authors want to explore the injuries it must be clear what is known about the strength influence in their occurrence, and studies that approached it.
The parts of the Discussion that is not related to the aim of the study has been removed, but the discussion overall has been extended.
On contrary, the obtained results are not enough discussed nor the study limitations referred. Please explain your results and compare them with those obtained by other authors.
A paragraph describing limitations has been included.
The conclusion must resume your study findings. The statement “Judo movement patterns are extremely demanding on the thigh muscles” is not a conclusion from your study that must be corrected.
The conclusion has been corrected.
Reviewer 3 Report
IJERPH- 1213668
Reviewer’s Comments
In the submitted manuscript, the authors study the isokinetic profile of judoists under the perspective of injury. A large cohort of elite Serbian female judo athletes were evaluated in a five repetition concentric-concentric knee extension/flexion maximal isokinetic test at a speed of 60degs/s for the mean, peak and normalized to body mass torque, as well as for the hamstring to quadriceps peak torque ratio for each lower limb. Higher absolute torque values were observed for the heavier athletes, but no differences were observed across weight categories when torque values were expressed relative to body weight. These results were revealed for both legs. Finally, a moderate negative correlation was revealed between hamstring to quadriceps peak torque ratio and years of training for both knee flexors and extensors.
The research is within the scope of the Journal. However, there are some issues that need to be addressed in order to consider this study for publication.
General comments
- Most of the Introduction and Discussion is about the relevance of the results of isokinetic test with injury prevention. No injury data were acquired in the study; thus, the Discussion is not fully supporting the findings of the study.
- A clear rational and hypothesis are not provided. The results of the isokinetic tests are checked for differences across the IJF categories; what is the reason for that and what did the authors expect to find? Also, provide the rationale to select the 60degs/s speed for the tests.
- The groups were defined according to IJF. Provide additional information concerning each group’s anthropometrics (age, body height).
- Methods: KER, KEL, KFR, KFL torque values are stated to be related to weight category (L82-83). Why was this normalization selected instead of each judoist’s body mass?
- Statistical analysis: No effect size measurements are mentioned. In addition, at some parts of the text (i.e. L103-105), a paired comparison is implied. Were there additional statistical tests run? The authors should also provide the rationale why the inter-limb differences were not checked.
- Discussion: Elaborate on the connection of the results of the study with the technical requirements and loading in judo. Also, provide a discussion on the findings of the correlation analysis.
- No limitations of the study are mentioned.
Specific comments
Abstract
- According to https://www.mdpi.com/journal/ijerph/instructions, the Abstract should follow the style of structured abstracts, but without headings. Thus, delete the numbers and corresponding headings in L16, L18, L19, L24 and L28.
Introduction
- L58-60: Provide a clear rationale of the study.
- L60: State a clear hypothesis.
Materials and Methods
- L70: Provide details about the ethical approval.
- L72: Propose to change to Experimental Procedure and Data Analysis.
- L75: Provide more information about the fixation of the participants onto the dynamometer and the calibration procedure. In addition, what was the consistency in terms of torque outcome across repetitions? Where there any criteria to exclude attempts from the analysis?
- L77: See the respective General Comment about the speed of the dynamometer.
- L77-78: Was there a random examination of which leg to test first?
- L80: Move Table 1 in the results section and provide details about the experimental parameters listed in the footnote (also add “torque” and “leg” when appropriate). See also the respective General Comment about the normalization data.
- Table 1: KER, KEL, KFR, KFL: Unit of measurement should be Nm/kg as in Figures 1 and 2.
- L92-98: See also the respective General Comment about the statistical tests and effect sizes.
- L93: ... IBM SPSS Statistics v23.0 (IBM Inc., Armonk, NY).
Results
- See the respective General Comments about the statistical analysis.
- L101: It is recommended to delete the first sentence.
- L106: “distant categories”: it is recommended to use other terms.
- L107-108: Mark the significant differences within the graph. Also, name the X axis categories after the IJF categories for a more clear depiction and understanding for the readers. The same comments for Figure 2.
- L108: …of extensors and flexors of the right knee. The same for L120.
- L120: Delete KEL.
- L124-125 & L134-135: Provide the symbols’ legend within the graph as well. As the graphs are in black and white, mark the regression lines in a different manner for a more clear depiction and understanding for the readers.
- L129-130: Use HQR only as it has been defined previously in the text.
Discussion
- See the respective General Comments about the Discussion and the Limitations of the Study.
- L141: Rewrite “demonstrated higher isokinetic is likely” (something is missing after isokinetic).
- L141: “…due to greater muscle mass”: As no body composition data were acquired, it is rather speculative; it is proposed to rewrite this providing further evidence or literature reference.
- L159: …11-12% of judo athletes.
- L159-163: This part of the text is not suitable for the smooth flow of the Discussion. It is suggested to be removed. See also the respective General Comments.
- L167: “The main muscles of the correct posture are the lower limbs”: Elaborate on this statement and its connection with the findings of the present study.
- L175: State the limitations of the study.
Conclusions
- L177-180: Expand on more practical implications for coaches and practitioners.
Author Response
R3
In the submitted manuscript, the authors study the isokinetic profile of judoists under the perspective of injury. A large cohort of elite Serbian female judo athletes were evaluated in a five repetition concentric-concentric knee extension/flexion maximal isokinetic test at a speed of 60degs/s for the mean, peak and normalized to body mass torque, as well as for the hamstring to quadriceps peak torque ratio for each lower limb. Higher absolute torque values were observed for the heavier athletes, but no differences were observed across weight categories when torque values were expressed relative to body weight. These results were revealed for both legs. Finally, a moderate negative correlation was revealed between hamstring to quadriceps peak torque ratio and years of training for both knee flexors and extensors.
The research is within the scope of the Journal. However, there are some issues that need to be addressed in order to consider this study for publication.
General comments
Most of the Introduction and Discussion is about the relevance of the results of isokinetic test with injury prevention. No injury data were acquired in the study; thus, the Discussion is not fully supporting the findings of the study.
The Discussion has been corrected.
A clear rational and hypothesis are not provided. The results of the isokinetic tests are checked for differences across the IJF categories; what is the reason for that and what did the authors expect to find? Also, provide the rationale to select the 60degs/s speed for the tests.
Groups have been formed base don the criterio developed by the IJF. Results have been presented base don the body weight of the participants on the day of the testing. Angular speed has been chosed base don the previous publications of similar nature.
The groups were defined according to IJF. Provide additional information concerning each group’s anthropometrics (age, body height).
The new table shows all of the required data.
Methods: KER, KEL, KFR, KFL torque values are stated to be related to weight category (L82-83). Why was this normalization selected instead of each judoist’s body mass?
Corrected. Thank you for the suggestion.
Statistical analysis: No effect size measurements are mentioned. In addition, at some parts of the text (i.e. L103-105), a paired comparison is implied. Were there additional statistical tests run? The authors should also provide the rationale why the inter-limb differences were not checked.
Thank you for the suggestion. We placed a particular emphasis on this part.
Discussion: Elaborate on the connection of the results of the study with the technical requirements and loading in judo. Also, provide a discussion on the findings of the correlation analysis.
Corrected.
No limitations of the study are mentioned.
The limitations section has been added.
Specific comments
Abstract
According to https://www.mdpi.com/journal/ijerph/instructions, the Abstract should follow the style of structured abstracts, but without headings. Thus, delete the numbers and corresponding headings in L16, L18, L19, L24 and L28.
Corrected.
Introduction
L58-60: Provide a clear rationale of the study.
Thank you for the comment. Corrected.
L60: State a clear hypothesis.
A hypothesis was developed.
Materials and Methods
L70: Provide details about the ethical approval.
Ethics approval has been added.
L72: Propose to change to Experimental Procedure and Data Analysis.
Corrected.
L75: Provide more information about the fixation of the participants onto the dynamometer and the calibration procedure. In addition, what was the consistency in terms of torque outcome across repetitions? Where there any criteria to exclude attempts from the analysis?
A paragraph has been added.
L77: See the respective General Comment about the speed of the dynamometer.
Explained.
L77-78: Was there a random examination of which leg to test first?
No.
L80: Move Table 1 in the results section and provide details about the experimental parameters listed in the footnote (also add “torque” and “leg” when appropriate). See also the respective General Comment about the normalization data.
The entire table has been corrected and added to the results section.
Table 1: KER, KEL, KFR, KFL: Unit of measurement should be Nm/kg as in Figures 1 and 2.
Thank you for the suggestion. The table has been corrected.
L92-98: See also the respective General Comment about the statistical tests and effect sizes.
Content added to the table.
L93: ... IBM SPSS Statistics v23.0 (IBM Inc., Armonk, NY).
Corrected.
Results
See the respective General Comments about the statistical analysis.
L101: It is recommended to delete the first sentence.
Deleted.
L106: “distant categories”: it is recommended to use other terms.
Corrected.
L107-108: Mark the significant differences within the graph. Also, name the X axis categories after the IJF categories for a clearer depiction and understanding for the readers. The same comments for Figure 2.
Weight classes have been marked. No significant difference was noted after normalizing values.
L108: …of extensors and flexors of the right knee. The same for L120.
Corrected.
L120: Delete KEL.
Deleted.
L124-125 & L134-135: Provide the symbols’ legend within the graph as well. As the graphs are in black and white, mark the regression lines in a different manner for a more clear depiction and understanding for the readers.
Marked.
L129-130: Use HQR only as it has been defined previously in the text.
Corrected.
Discussion
See the respective General Comments about the Discussion and the Limitations of the Study.
L141: Rewrite “demonstrated higher isokinetic is likely” (something is missing after isokinetic).
Corrected.
L141: “…due to greater muscle mass”: As no body composition data were acquired, it is rather speculative; it is proposed to rewrite this providing further evidence or literature reference.
Corrected.
L159: …11-12% of judo athletes.
Corrected.
L159-163: This part of the text is not suitable for the smooth flow of the Discussion. It is suggested to be removed. See also the respective General Comments.
Removed.
L167: “The main muscles of the correct posture are the lower limbs”: Elaborate on this statement and its connection with the findings of the present study.
Corrected.
L175: State the limitations of the study.
A paragraph has been added.
Conclusions
L177-180: Expand on more practical implications for coaches and practitioners.
A paragraph has been added.
Round 2
Reviewer 1 Report
Congratulations! The authors improved the manuscript. I agree with this version
Author Response
R1
Congratulations! The authors improved the manuscript. I agree with this version
Thank you for your valuable input!
Reviewer 2 Report
The results of the study performed by the authors are not enough discussed nor compared with those obtained by other authors. Please do not confound power with strength.
The conclusion must resume the study findings. It should answer the study aim.
Author Response
R2
The results of the study performed by the authors are not enough discussed nor compared with those obtained by other authors. Please do not confound power with strength.
The conclusion must resume the study findings. It should answer the study aim.
The discussion and conclusion were corrected.
Reviewer 3 Report
IJERPH- 1213668 –v2
Reviewer 3 Comments
In the resubmitted manuscript, the authors did extensive modification according to the reviewers’ comments. Nevertheless, there is still a number of topics that need to be addressed.
General comments
- Rational. More information is needed in the Introduction to establish the necessity of the research concerning the inter-category comparison.
- The authors should also provide the rationale why the inter-limb differences were not checked. Inter-limb asymmetry and torque imbalances are also causal factors for injury, as stated in L57-59.
- Statistical analysis: Effect size measurements are now mentioned but not described either in the statistical analysis subsection (2.3) or in the results (i.e. its magnitude as large, trivial, small effect).
- Discussion: In the resubmitted version, no discussion on the findings of the correlation analysis is provided and the relevance of the isokinetic torque analysis findings with injury prevention for the athletes measured.
Specific comments
Title
- Consider including “Serbian” and the study design (i.e. of different weight category).
Abstract
- L19: knee flexors and extensors isokinetic torque analysis (instead of quadriceps and hamstring) as in L22-23 (again, quadriceps and hamstring should be deleted).
Introduction
- L50: Isokinetic torque rather than strength.
- L64-65: peak isokinetic muscle torques.
Materials and Methods
- According to the comments on the initial submission, the following remain to be addressed: 1) L75: What was the consistency in terms of torque outcome across repetitions? Where there any criteria to exclude attempts from the analysis = namely: the intra-trial reliability among the five repetitions and a threshold not to include an attempt in the analysis, 2) L77-78: Was there a random examination of which leg to test first? = include your response in the text and provide evidence/rationale, 3) L92-98: See also the respective General Comment about the statistical tests and effect sizes = partial eta squared is used, but is should be also stated in the Statistical Analysis subsection, along with the thresholds to define the effect size as trivial, small, medium, high, etc. (2.3, L103-108)
- L78-81: Rephrase as “approved” is mentioned twice.
- L88: seat adjustment according to weight? Do you mean the inertia correction?
- L94-95: It is recommended to rephrase as: “The test velocity was used according to previous studies [6,16]“.
- L98: …(in Nm)…
- L98: Delete “(hamstring)”.
- L99: Delete (“quadriceps”).
- L99: Is the abbreviation for the hamstring to quadriceps ratio H/Q should be defined as HQR and HQL here, as H/Q is not used again contrarily to both HQR and HQL.
Results
- See the respective General Comments about mentioning the effect sizes.
- L116: See the Specific Comment for L99.
- L120: State that ± corresponds for Standard Deviation.
- L120: Table 1: Weight for -52kg category is 52.00±1.80 kg; were there athletes heavier than 52kg?
- L143: As no intra-limb statistical comparison was done for peak flexion and extension torque, use “rather than” instead of “compared”.
- L144: report all p values with 3 decimal digits.
- L153: delete “that”.
- L158: report all p values with 3 decimal digits.
- L160: Figure 3: insert legend within the graph.
- L164: report all p values with 3 decimal digits.
- L166: Figure 4: insert legend within the graph.
Discussion
- According to the comments on the initial submission, the following remain to be addressed: 1) L141: Rewrite “demonstrated higher isokinetic is likely” (something is missing after isokinetic) = higher isokinetic muscle torque, 2) L141: “…due to greater muscle mass”: As no body composition data were acquired, it is rather speculative; it is proposed to rewrite this providing further evidence or literature reference.
- L192: …isokinetic torque…
- L193: It is suggested to use “discussed” instead of “identified”.
- L202-203: Elaborate on this statement and its connection with the findings of the present study.
- L203: See the respective General Comments about the results of the correlation analysis.
References
- L260: Isokinetic Dynamometry: Applications and Limitations.
Author Response
R3
In the resubmitted manuscript, the authors did extensive modification according to the reviewers’ comments. Nevertheless, there is still a number of topics that need to be addressed.
General comments
- Rational. More information is needed in the Introduction to establish the necessity of the research concerning the inter-category comparison.
The study was conducted to additionally develop classificatory standards in peak isokinetic muscle torques of the knee joint extensors and flexors for female judo athletes.
- The authors should also provide the rationale why the inter-limb differences were not checked. Inter-limb asymmetry and torque imbalances are also causal factors for injury, as stated in L57-59.
Corrected (L 163-166).
- Statistical analysis: Effect size measurements are now mentioned but not described either in the statistical analysis subsection (2.3) or in the results (i.e. its magnitude as large, trivial, small effect).
Corrected.
- Discussion: In the resubmitted version, no discussion on the findings of the correlation analysis is provided and the relevance of the isokinetic torque analysis findings with injury prevention for the athletes measured.
Corrected.
Specific comments
Title
- Consider including “Serbian” and the study design (i.e. of different weight category).
Corrected.
Abstract
- L19: knee flexors and extensors isokinetic torque analysis (instead of quadriceps and hamstring) as in L22-23 (again, quadriceps and hamstring should be deleted).
Corrected.
Introduction
- L50: Isokinetic torque rather than strength.
Corrected.
- L64-65: peak isokinetic muscle torques.
Corrected.
Materials and Methods
- According to the comments on the initial submission, the following remain to be addressed: 1) L75: What was the consistency in terms of torque outcome across repetitions? Where there any criteria to exclude attempts from the analysis = namely: the intra-trial reliability among the five repetitions and a threshold not to include an attempt in the analysis, 2) L77-78: Was there a random examination of which leg to test first? = include your response in the text and provide evidence/rationale, 3) L92-98: See also the respective General Comment about the statistical tests and effect sizes = partial eta squared is used, but is should be also stated in the Statistical Analysis subsection, along with the thresholds to define the effect size as trivial, small, medium, high, etc. (2.3, L103-108)
Maximal torque values were recorded out of five repetitions.
The dominant leg was tested first.
Mentioned in the statistical analysis section.
- L78-81: Rephrase as “approved” is mentioned twice.
Corrected.
- L88: seat adjustment according to weight? Do you mean the inertia correction?
Yes
- L94-95: It is recommended to rephrase as: “The test velocity was used according to previous studies [6,16]“.
Corrected.
- L98: …(in Nm)…
Corrected.
- L98: Delete “(hamstring)”.
Corrected.
- L99: Delete (“quadriceps”).
Corrected.
- L99: Is the abbreviation for the hamstring to quadriceps ratio H/Q should be defined as HQR and HQL here, as H/Q is not used again contrarily to both HQR and HQL.
Corrected.
Results
- See the respective General Comments about mentioning the effect sizes.
- L116: See the Specific Comment for L99.
Corrected.
- L120: State that ± corresponds for Standard Deviation.
Corrected.
- L120: Table 1: Weight for -52kg category is 52.00±1.80 kg; were there athletes heavier than 52kg?
There were, but they were from the same category.
- L143: As no intra-limb statistical comparison was done for peak flexion and extension torque, use “rather than” instead of “compared”.
Corrected.
- L144: report all p values with 3 decimal digits.
Corrected.
- L153: delete “that”.
Corrected.
- L158: report all p values with 3 decimal digits.
Corrected.
- L160: Figure 3: insert legend within the graph.
Corrected.
- L164: report all p values with 3 decimal digits.
Corrected.
- L166: Figure 4: insert legend within the graph.
Corrected.
Discussion
- According to the comments on the initial submission, the following remain to be addressed: 1) L141: Rewrite “demonstrated higher isokinetic is likely” (something is missing after isokinetic) = higher isokinetic muscle torque, 2) L141: “…due to greater muscle mass”: As no body composition data were acquired, it is rather speculative; it is proposed to rewrite this providing further evidence or literature reference.
Corrected.
- L192: …isokinetic torque…
Corrected.
- L193: It is suggested to use “discussed” instead of “identified”.
Corrected.
- L202-203: Elaborate on this statement and its connection with the findings of the present study.
Corrected.
- L203: See the respective General Comments about the results of the correlation analysis.
Corrected.
References
- L260: Isokinetic Dynamometry: Applications and Limitations.
Corrected.